# Is Agriculture Always a GHG Emitter? A Combination of Eddy Covariance and Life Cycle Assessment Approaches to Calculate C Intake and Uptake in a Kiwifruit Orchard

**Federica Rossi, Camilla Chieco, Nicola Di Virgilio, Teodoro Georgiadis**  **and Marianna Nardino** *

Institute of BioEconomy, National Research Council, 40129 Bologna, Italy; Federica.Rossi@ibe.cnr.it (F.R.);
Camilla.Chieco@ibe.cnr.it (C.C.); nicola.divirgilio@ibe.cnr.it (N.D.V.); Teodoro.Georgiadis@ibe.cnr.it (T.G.)
* Correspondence: Marianna.Nardino@ibe.cnr.it; Tel.: +39-051-639-9001

**Abstract:** While a substantial reduction of GHG (greenhouse gases) is urged, large-scale mitigation implies a detailed and holistic knowledge on the role of specific cropping systems, including the effect of management choices and local factors on the final balance between emissions and removals, this last typical of cropping systems. Here, a conventionally managed irrigated kiwifruit orchard has been studied to assess its greenhouse gases emissions and removals to determine its potential action as a C sink or, alternately, as a C source. The paper integrates two independent approaches. Biological $CO_2$ fluxes have been monitored during 2012 using the micrometeorological Eddy covariance technique, while life cycle assessment quantified emissions derived from the energy and material used. In a climatic-standard year, total GHG emitted as consequence of the management were 4.25 t $CO_2$-eq$^{-1}$ ha$^{-1}$ yr$^{-1}$ while the net uptake measured during the active vegetation phase was as high as 4.9 t $CO_2$ ha$^{-1}$ yr$^{-1}$. This led to a positive contribution of the crop to $CO_2$ absorption, with a 1.15 efficiency ratio (sink-source factor defined as t $CO_2$ stored/t $CO_2$ emitted). The mitigating activity, however, completely reversed under extremely unfavorable climatic conditions, such as those recorded in 2003, when the efficiency ratio became 0.91, demonstrating that the occurrence of hotter and drier conditions are able to compromise the capability of *Actinidia* to offset the GHG emissions, also under appropriate irrigation.

**Keywords:** carbon balance; mitigation; emissions; efficiency ratio; irrigated agriculture; life cycle assessment; *Actinidia deliciosa*

## 1. Introduction

A growing consciousness about the opportunity to promote farm management actions able to preserve environmental, landscape and social values in the rural society has been emerging in the last decade [1–4], in parallel with the shortage of non-renewable resources and climate change issues. Agriculture is now strongly solicited to move towards a climate smart–based strategy [5] addressing sustainability, adaptation, and mitigation while aligned to MDGs of climate action [6].

Modern agriculture is claimed as a heavy resource-user and high-impact activity. The literature reports that the food system contributes to 20–29% of the total global anthropogenic GHG emissions, 80–86% of this coming from primary production (with major differences between Countries), while the remaining is due to upstream and downstream pre- and post-production chain [7–10]. A food system approach is now correctly proposed [11] for the implementation of climate change adaptation and mitigation measures, as fundamental connections between field production and consumer demand may facilitate the design of integrated solutions.

In its new climate law, EU committed to reach net neutrality in 2050. While a substantial decarbonization is urged in all sectors, including agriculture, it also recognizes the significant role of the land sector in increasing the EU sink, necessary to compensate

residual emissions in 2050. This implies a detailed knowledge on the potential contribution that each single land use system, including orchards, can deliver in terms of carbon removal and net sequestration, under different management choices and climate. Even if emissions from agriculture can still be reduced, its reduction potential is lower than other sectors, very costly and cannot be reduced to zero. This means that most probably, as also suggested by several models run at EU level (European Commission—stepping up Europe's 2030 Climate Ambition, 2020) residual emissions in 2050 will mostly be emissions from agriculture. It would indeed make sense to compare for each land use system the ratio between the sink they can provide and emissions due to associated management in order to understand their potential to deliver net neutrality.

Each agricultural system indeed acts differently in terms of $CO_2$ equivalent exchanges, as there is a specific balance between physiological crop activities (photosynthesis, respiration, and soil contribution) and direct and indirect emissions due to the use of input in the farm management. Among most valuable crops, fruit tree orchards are intensively managed, exchanging $CO_2$ for a life-period of at least 20–25 consecutive years. Based on the few long-term $CO_2$ fluxes observations, an active annual C sink activity has been reported so far only for some fruit tree species. Rossi et al. [12] observed in a kiwifruit orchard a NEE (Net Ecosystem Exchange) of 3.09 t $CO_2$ ha$^{-1}$ yr$^{-1}$ during the spring-to-harvest dry, hot 2003, and two-years measurements in an intensive managed olive reflected an NEE dependency on different yearly meteorological conditions (13.5 and 11.6 t $CO_2$ ha$^{-1}$ yr$^{-1}$) [13]. Marras et al. [14] found that a mature Mediterranean vineyard was able to sequester annually 7.15 t $CO_2$ ha$^{-1}$ yr$^{-1}$, with a large monthly variability. Here, a kiwifruit orchard conventionally managed has been studied to assess its greenhouse gases annual exchange and determine its potential action as a sink or, alternately, as a source. Kiwifruit has been chosen among other species considering the actual, and the expected, cultivation extensions. This perennial climbing species is spread in many Asian, European, Australian, and American countries due its tolerating continental and maritime climatic conditions and has therefore a significant economic importance. Kiwi is considered a superfood due to its chemical composition, high vitamin and phenols, low caloric content. Studies confirmed several benefits associated with its consumption, such as the reduction of oxidative stress and protection against heart disease, cancer, diabetes, vascular and central nervous system diseases. Actinidin is also a potential enzyme that may help to hydrolyze proteins, including gluten, and may effectively supplement celiac diets [15]. Last, but not least, peels, seeds and leaves or pruning remaining from processing may be used to recover target bioactive components with potential high commercial applications. encouraging the "zero waste" principle [16]. The recognition of nutritional/functional natural properties leverages a progressive increase of the acreage dedicated to this crop raising the significance of its effective role as contributor to GHG emission, or/and its efficiency in sequestering carbon.

While accurate micrometeorological application permits the assessment of the dynamics of carbon cycle in forests and natural ecosystems, the limited orchard extensions constrained the same measurements in agricultural tree crop extensions providing little information about their efficiency.

This paper integrated two independent approaches. Biological $CO_2$ fluxes have been monitored during using the micrometeorological Eddy Covariance technique (EC) in 2012, while Life Cycle Assessment (LCA) was used to quantify emissions derived from the energy and material used in the temporal boundaries in the same period of the same year. The use of kg as the functional unit in LCA and the very similar crop management in the two fields allowed to extrapolate the LCA results to a very different climatic year (2003), in which C fluxes had already been published [12]. A comparison of the efficiency of the orchard as a sink between the years was then made possible and is reported in this paper.

## 2. Materials and Methods

### 2.1. Orchard Description

Measurements were carried out in 2012 on a conventionally managed 10-year-old kiwifruit orchard (*Actinidia deliciosa* var *deliciosa* cv Hayward) located in a flat area of the Po plane located between Faenza and Ravenna (44°20′39″ N, 11°59′02″ E, 15 m a.s.l), Emilia Romagna, Italy. The soil texture was sand 50%, silt 26%, clay 24%. The volumetric water content (VWC) at field capacity was 29.14%, wilting point was 13.66% VWC and available water 15.48% VWC. Soil organic matter was 1.32% of total topsoil.

Plants were trained at a T shape and spaced $3 \times 5$ m apart (670 plants ha$^{-1}$). The staminate "Tomuri" selection was used as a pollinizer (with a ratio females/male 9:1).

Kiwifruit is a highly water demanding crop, therefore irrigation is mandatory in almost all its growing zones. The irrigation system was equipped with four micro sprinklers (4 L h$^{-1}$) per tree. During the period of active growth (June to September) the total volume of water supplied was about 3500 m$^3$ ha$^{-1}$, with a daily rate depending on the monthly weather conditions. Irrigation plus precipitation always well compensated the evapotranspiration losses directly quantified from measured latent heat flux via Eddy covariance (Table 1).

**Table 1.** Water balance components during the summer when irrigation was supplied.

|  | June | July | August | September | October | Total |
|---|---|---|---|---|---|---|
| Evapotranspiration (mm$^3$ ha$^{-1}$) | 1088 | 1247 | 1204 | 892 | 538 | 4969 |
| Irrigation (mm$^3$ ha$^{-1}$) | 657 | 1081 | 1038 | 514 | 201 | 3491 |
| Precipitation (mm$^3$ ha$^{-1}$) | 624 | 144 | 289 | 296 | 318 | 1671 |
| Irrigation + Precipitation (mm$^3$ ha$^{-1}$) | 1281 | 1225 | 1327 | 810 | 519 | 5162 |

The leaf area of a representative sample (around 20 vines) was measured at weekly intervals during the growing season up to middle July, when new leaves formation and current leaf expansion ceased. The leafy area of each kiwifruit plant was quickly increasing, reaching values up to 25 m$^2$ plant$^{-1}$ in the first two weeks of the season, with final values up to 37 m$^2$ plant$^{-1}$ at the beginning of July. In total, the actively exchanging orchard leafy surface was around 25,000 m$^2$ ha$^{-1}$. Temporal increase of the leaf area index (LAI) (m$^2$ leaf area/m$^{-2}$ allotted ground area) was directly calculated by leaf number, their average surface area, and the canopy projection on the ground of the canopy area. LAI at the end of the growing phase was around 3. Yield was 23.9 t ha$^{-1}$.

### 2.2. The Vegetation $CO_2$ Fluxes: Eddy Covariance

The orchard extension was 10.2 ha. Given the limitations in finding homogenous surface conditions as often happens in many rural landscapes, a major concern was to assess how a non-homogeneous source-sink distribution could have partly conditioned a correct flux monitoring. The location of the EC tower was then chosen after a preliminary field analysis based on the five-years records of wind velocity and wind direction at the farm (Figure 1).

During the spring-summer vegetative season the prevalent wind directions were east and south/west, and the greatest percentage of data occurred for wind speed lower than 3 m/s. The location of the EC tower, shown in Figure 2 with the white dot, was consequently chosen to collect the highest number of upwind streams coming from the kiwifruit orchard. The footprint analysis was post processed after the period of measurements following the methodology proposed by Schuepp et al. [17]. Figure 2 shows the upwind distances most likely contributing to the maximum total flux: the data collected are, in the greatest part, within the orchard surface and the fetch conditions were considered acceptable for a reliable flux data collection.

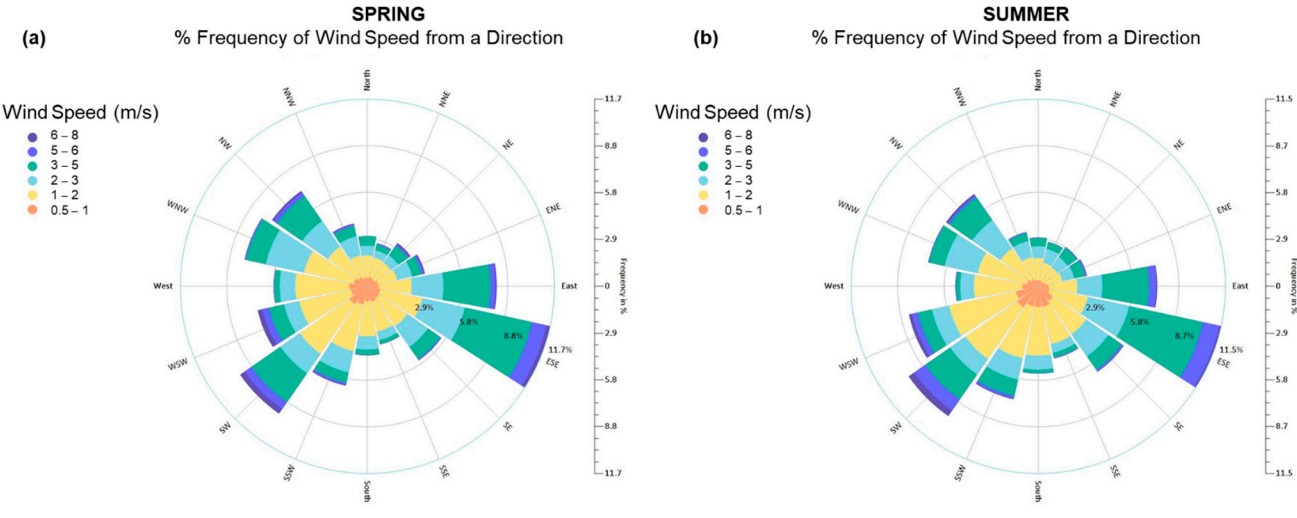

**Figure 1.** Frequency of wind speed (m s$^{-1}$) from wind direction sectors recorded during 5 years in (**a**) spring and (**b**) summer.

$CO_2$ fluxes and energy balance components were measured at 3.5 m (about two times the canopy height) using a three-dimensional sonic anemometer (Metek GmbH, USA1, Elmshorn, Germany) and an open path infra-red absorption gas analyzer (IRGA-Li7500, LiCor Inc., Lincoln, NE, USA) both positioned on the top of the tower. Wind, water vapour, carbon dioxide and temperature were sampled at 10 Hz.

On the same tower, a CNR1 net radiometer (Kipp and Zonen, Delft, The Netherlands) measured the net available energy at the surface. Three flux plates (HFP01, Campbell Sci., installed at a depth of 5 cm in three different points at the same distance from the girth of the tower measured the soil heat flux. All data were stored on a CR1000 (Campbell Sci., USA) data logger.

EC data processing followed the guidelines of the standard EUROFLUX methodology [18]. The half-hour mean flux values were post processed through the calculation of mean covariances between w' and the scalars c', q' and T' (where w is the vertical component of the wind speed vector, c' is the $CO_2$ concentration, q' is the $H_2O$ concentration and T' is the air temperature).

The applicability of EC is restricted by several assumptions: horizontal homogeneity of the upwind surface, homogeneity of the turbulence and mean flow, stationarity, storage, sensors misalignments, changes in air density, etc. [19,20]. During the post processing a quality check was applied together with specific routines to remove the common errors: running means to avoid de-trending problems, three angle coordinate rotations of the wind vector to remove the effects of instrument tilt or terrain irregularity on the airflow, de-spiking, and stationarity.

Surface energy balance closure is the test enabling to evaluate the good quality of EC data. The energy balance closure, obtained by plotting net radiation (Rn) minus the soil heat flux (G) against the sum of sensible heat flux (H) and latent heat flux (LE), was verified considering half-hour averaged values during the entire measurement season (Figure 3). The observed energy balance closure was less than 20% (correlation coefficients: $R^2 = 0.89$, slope = 0.82) and gave us confidence about the accuracy of the measurements.

The storage of $CO_2$ in the layer below the canopy during the nocturnal stable atmospheric conditions was estimated by using the time-change in the $CO_2$ concentration measured at the top of the mast [21]:

$$F\Delta S = \frac{\Delta C(z)}{\Delta t}\Delta z \tag{1}$$

where $\Delta C(z)$ is the change in $CO_2$ at the height $z$, $\Delta t$ is the measurements time period, and $\Delta z$ the height of the layer.

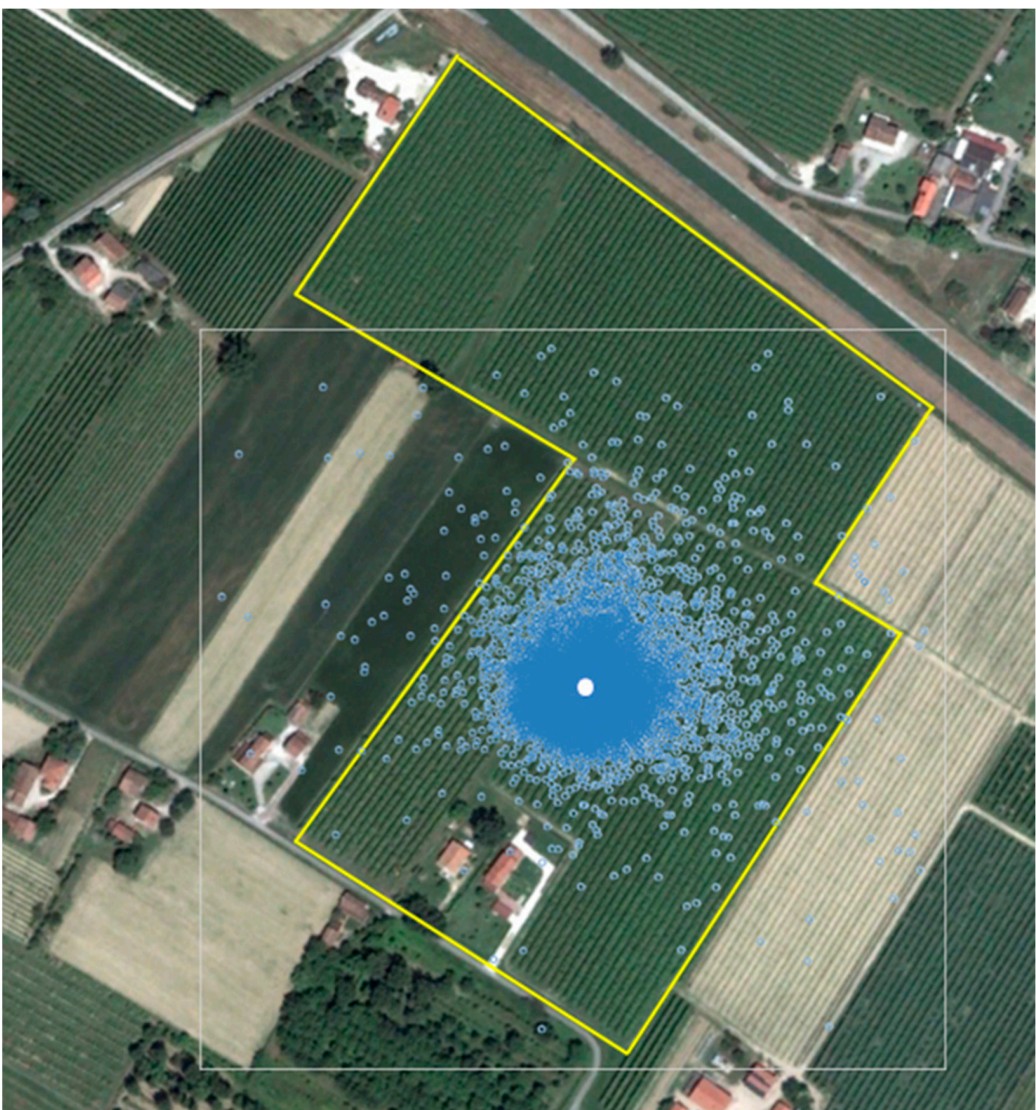

**Figure 2.** Map of the area from Google Earth. The square encloses the kiwi surfaces, and the white dot indicates the location of the EC tower. Blue dots show the maximum of total flux obtained by footprint analysis.

To avoid the underestimation of the nocturnal surface exchanges [22,23], night data were inspected to detect invalid values (i.e., negative values of carbon dioxide flux, values of respiration outside of the trend). Such a procedure was preferred to the more usually adopted correction based on established threshold u* value because of the regular structure of the orchard and the sparse arrangement of trees [13].

Gaps due to some unfavorable micro-meteorological conditions, instrument failure and data quality check led to a data coverage of 61% of the whole period.

To obtain daily, monthly, or seasonal integrated balances, a standardized gap filling was applied. The methodology adopted here consisted of introducing our data in the on-line Eddy covariance gap-filling and flux partitioning tools [24,25] adopted by several European flux network sites.

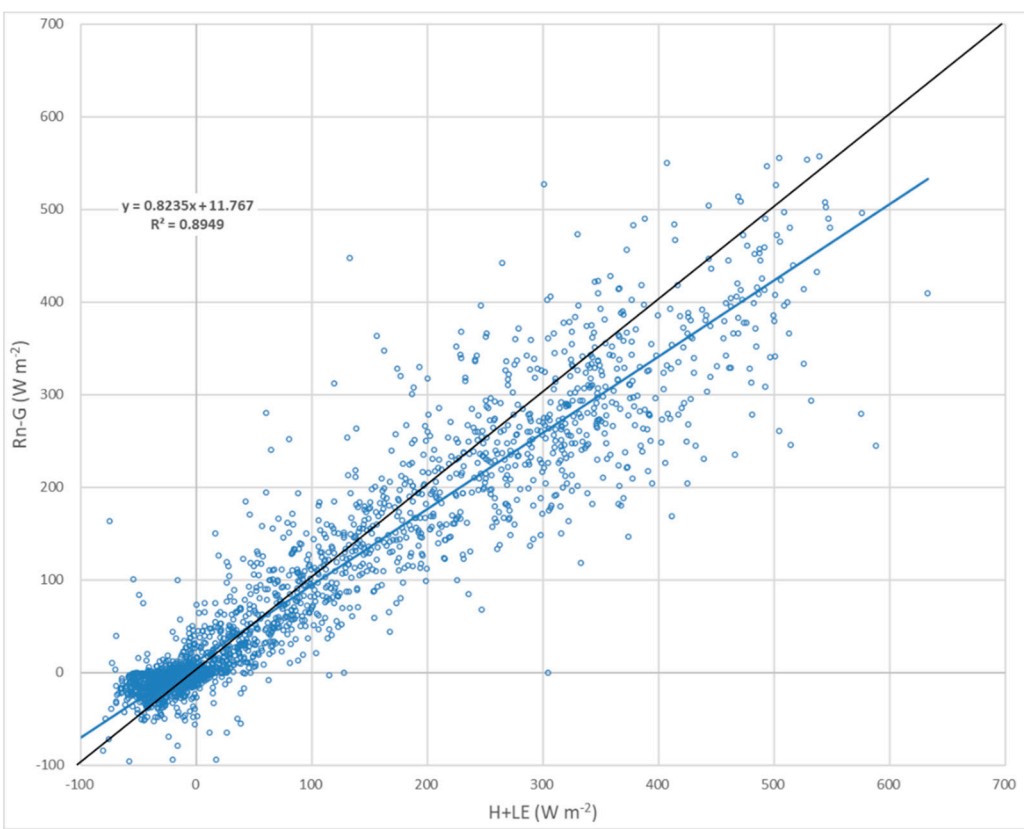

**Figure 3.** Energy balance closure. H is the sensible heat flux, LE is the latent heat flux, Rn is the net radiation and G is the soil heat flux.

Total carbon exchange terms were estimated at a monthly scale, and the net ecosystem exchange (NEE) was computed from the complete series of $CO_2$ fluxes (measured and gap-filled). In accordance with the atmospheric science convention, positive values represent a release from the surface (upward fluxes) and negative values represent an uptake from the surface (downward fluxes). The ecosystem respiration (ER) was estimated by summing the quality-controlled night-time ecosystem respiration data and the night values simulated when missing. The gross primary productivity (GPP) for each 30 min period was the arithmetic sum between NEE and ER.

### 2.3. CO₂-Eq Emissions by Orchard Management: Life Cycle Assessment

When managing agricultural systems, other GHGs than $CO_2$ are emitted. The evaluation of $CO_2$-eq emissions deriving from the seasonal orchard management was analytically defined through the quantification of all the inputs introduced within the system from the farmer, including emission-associated machineries. Different GHGs have a warming influence on the global climate system depending on their radiative properties and lifetimes in the atmosphere. $CO_2$-eq have been obtained by multiplying the emission of specific different GHGs by their Global Warming Potential (GWP) for the given time horizon, and were assumed as the measurement unit, as suggested by IPCC [26].

The orchard system was described based on primary data directly gathered from the kiwifruit growers with a structured interview, set up to acquire details on the practices and the materials used, calendars of key operations including various related inputs, irrigation system and water scheduling, etc. Inputs and materials regarding the orchard establishment phase were also collected and separately analyzed.

The agronomic practices adopted were in line with those typically embraced by local farmers, as checked by local cooperative technicians, and were therefore assumed as representatives of a conventional kiwifruit orchard management in a standard production

district. The same results were then utilized to infer the emission trend during the 2003 vegetative season.

LCA calculated Emissions of GHGs (as $CO_2$-eq) were calculated following a life cycle assessment approach in which both direct and input-embedded upstream emissions were taken into consideration, considering as boarder of the system under assessment the farm gate, including fruit harvesting. SimaPro 7.0 software (PRé Consultants, Amersfoort, The Netherlands), with the widespread dataset Ecoinvent 1.1 (Swiss Centre for Life Cycle Inventory, Zurich, Switzerland) and LCA Food DK, were adopted to model and analyze the orchard. Impact evaluation method was CML 2 baseline 2001, which elaborates on the problem-oriented (midpoint) approach. The CML Guide provides a list of impact assessment categories in which, among others, the global warming 100a, expressed as kg $CO_2$-eq, is released.

Since the EC tower was measuring only $CO_2$ fluxes, the $CO_2$-eq related to $N_2O$ emissions from the soil through denitrification processes were estimated in relation to the fertilization inputs using IPCC emission factors [27]. Our estimation based on the IPCC relation of $N_2O$ emission (1% of external-applied N), led to a value of 722.81 kg $CO_2$-eq $ha^{-1}$ $yr^{-1}$, consistent with the literature. In the work carried out in New Zealand, the value of $CO_2$-eq deriving from $N_2O$ emissions from soil were estimated with the use of Overseer model and resulted to be 249 kg $CO_2$-eq $ha^{-1}$ $yr^{-1}$ [3].

Foster and Matthew [28] considered insignificant the weight of the contribution related to the production phases of such inputs as tractors, machineries, fertilizers, particularly when split among the various field operations and the lifetime of the orchard. We however chose to also include these inputs into the calculation, to be as precise as possible. Emission values for tractors and implements were allocated in proportion to the number of hours dedicated to the various operations, such as mowing, spraying and harvesting with respect to the working life of each tractor and implements.

The GHG emissions related to the establishment phase of the orchard, including first deep ploughing, were considered once during the 15 years orchard's lifetime.

The analysis was carried out using one kg of kiwi yield as a functional unit.

## 3. Results

### 3.1. $CO_2$ Uptake from Eddy Covariance Measurements

The use of the micrometeorological EC technique avoided, in our case, the use of modeling to estimate each C soil emission components, directly providing the result of the balance among photosynthesis, plant respiration and soil emissions, and thus largely increasing the reliability of values.

Figure 4 reports the "typical day" energy and the $CO_2$ fluxes computed for a summer and a winter month as the hourly averages of the values measured in all days at the same hour. The $CO_2$ flux from and to the canopy followed the typical seasonal trend over broadleaf species: during the vegetative season, when leaves are present, the $CO_2$ uptake is higher than the $CO_2$ release through respiration. During the winter, the ecosystem respiration prevails due to the absence of vegetation, and $CO_2$ uptake is totally missing.

During the summer, the orchard reached its maximum $CO_2$ fixation at about 4.3 mol $m^{-2}$ $day^{-1}$ $CO_2$ (Figure 4a), when canopy active fixation patterns were positive during the day and negative at night, with a turn at around 7 am and 5 pm, in agreement with Buwalda et al. [29] who measured kiwi $CO_2$ net assimilation enclosing vines in whole-canopy cuvettes, showing that $CO_2$ fixation was lower in the afternoon than in the morning at any PAR (Photosynthetically Active Radiation) value. During the winter, the orchard acted as a source and released about 0.5 mol $m^{-2}$ $day^{-1}$ $CO_2$ (Figure 4b). The amount of energy available for all surface exchange processes, such as latent heat flux and sensible heat, were much lower than in the summer season (150 W $m^{-2}$ in July against 700 W $m^{-2}$ in November), resulting in a reduced exchange of all turbulent flows, including $CO_2$ flux concomitant to leaf fall. At this time, the orchard turned to become a $CO_2$ source rather than a sink. A large amount of $CO_2$ came either from soil or plant respiration (75% of the

total $CO_2$), as already observed for the same crop, for which similar percentage was found (Rossi et al., 2007).

Figure 5 details the monthly cumulated NEE, GPP and RE. June and July were the more active $CO_2$ uptake months, with the highest GPP reached in July.

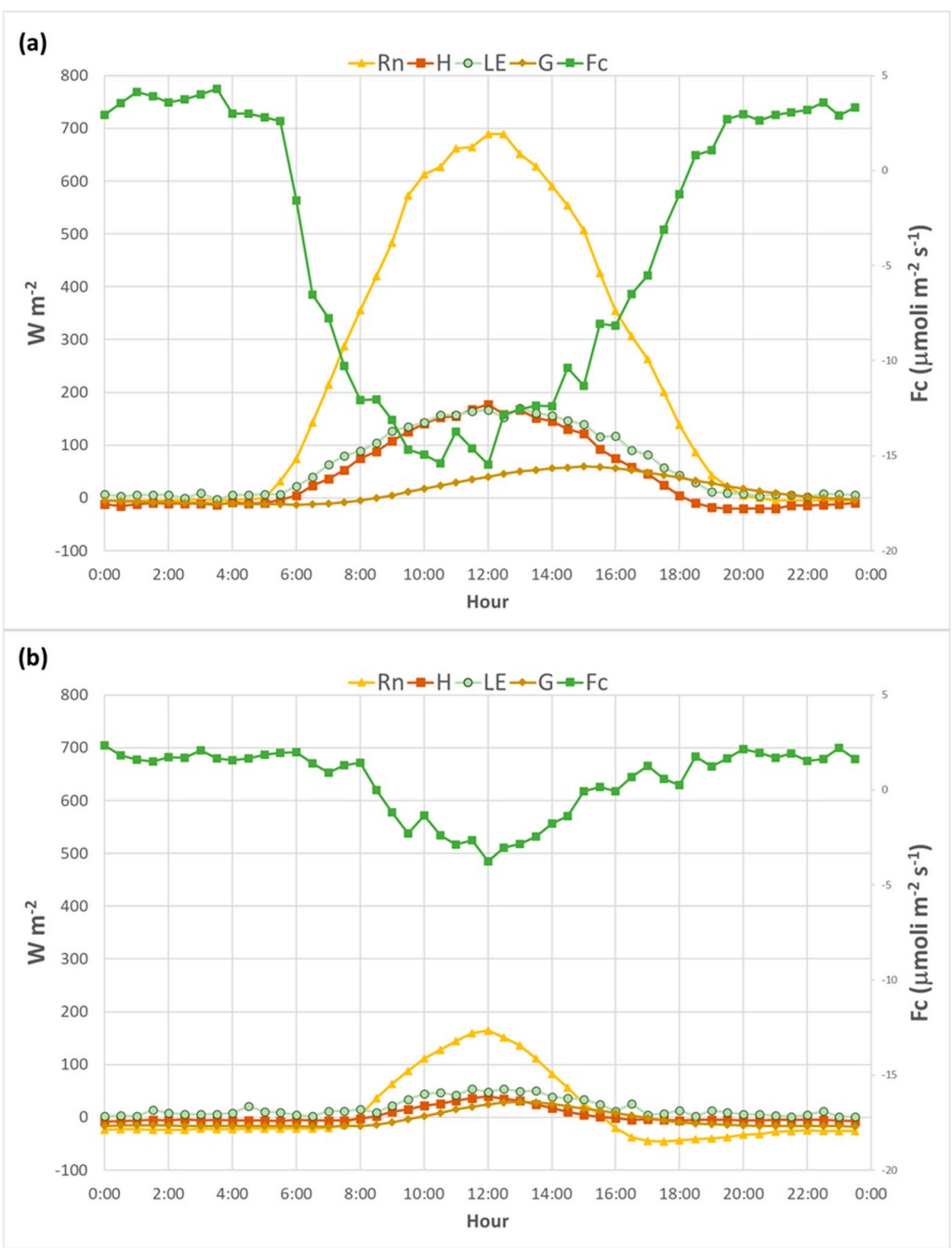

**Figure 4.** Surface energy balance components (H is sensible heat flux, LE is latent heat flux, G is soil heat flux, Rn is net radiation) and $CO_2$ flux (Fc) in (**a**) a typical summer-July-day and (**b**) a winter-November-day.

When comparing the results with those obtained in 2003 in a nearby kiwifruit orchard having comparable extension that was similarly managed (Figure 6), a much higher carbon sequestration was recorded during the vegetative season (6 months) of the current 2012 (NEE 4.9 t C ha$^{-1}$ against 3.2 t in 2003, GPP 13.6 t C ha$^{-1}$ in 2012 and 10.9 t C ha$^{-1}$ in 2003). As the two orchards were comparable for geographic location and field arrangement, a

strong dependency of C exchanges on climate conditions may be inferred. In 2003 the strongest heat wave of the last 20 years was recorded in Europe, and the combined effect of drought and high temperatures led to extensive damages and losses of agricultural productivity [30].

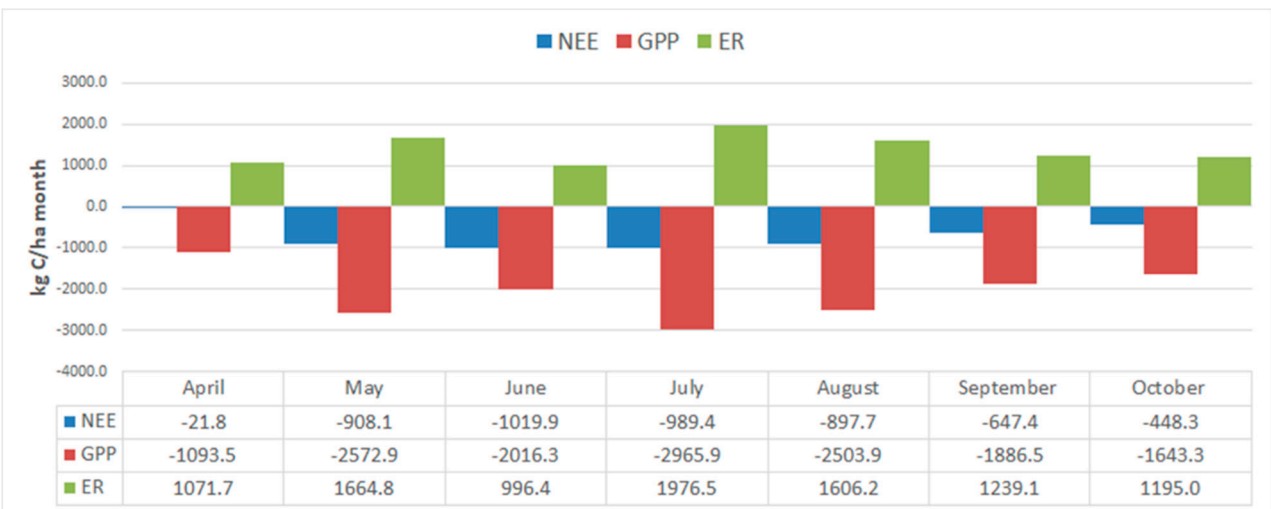

**Figure 5.** Monthly cumulated net ecosystem exchange (NEE), gross primary production (GPP) and ecosystem respiration (RE) during the vegetative season.

In the first part of the 2003 vegetative season (May and June), the total absence of precipitation added to seasonal temperatures higher than in 2012 (Figure 7), even if the orchard was irrigated to compensate evapotranspiration losses (Table 2), reflected into lower NEE and GPP values.

**Table 2.** NEE, GPP and ER measured through eddy covariance.

|  | NEE <br> (kg $CO_2$ $ha^{-1}$ $Month^{-1}$) | GPP <br> (kg $CO_2$ $ha^{-1}$ $Month^{-1}$) | ER <br> (kg $CO_2$ $ha^{-1}$ $Month^{-1}$) |
|---|---|---|---|
| January | 707 | −741 | 1448 |
| February | 713 | −1345 | 2058 |
| March | 720 | −1950 | 2670 |
| April | −80 | −4010 | 3930 |
| May | −3330 | −9435 | 6105 |
| June | −3740 | −7394 | 3654 |
| July | −3628 | −10,875 | 7248 |
| August | −3292 | −9182 | 5890 |
| September | −2374 | −6919 | 4544 |
| October | −1644 | −6026 | 4382 |
| November | 870 | −1450 | 2320 |
| December | 640 | −830 | 1470 |
| Total | −14,438 | −60,157 | 45,719 |

Photosynthetic data in 2003 showed a premature stomata closure after 10 am due to excessive heat [12], while in 2012 the maximum was around midday, when the PAR reached its maximum value (Figure 4).

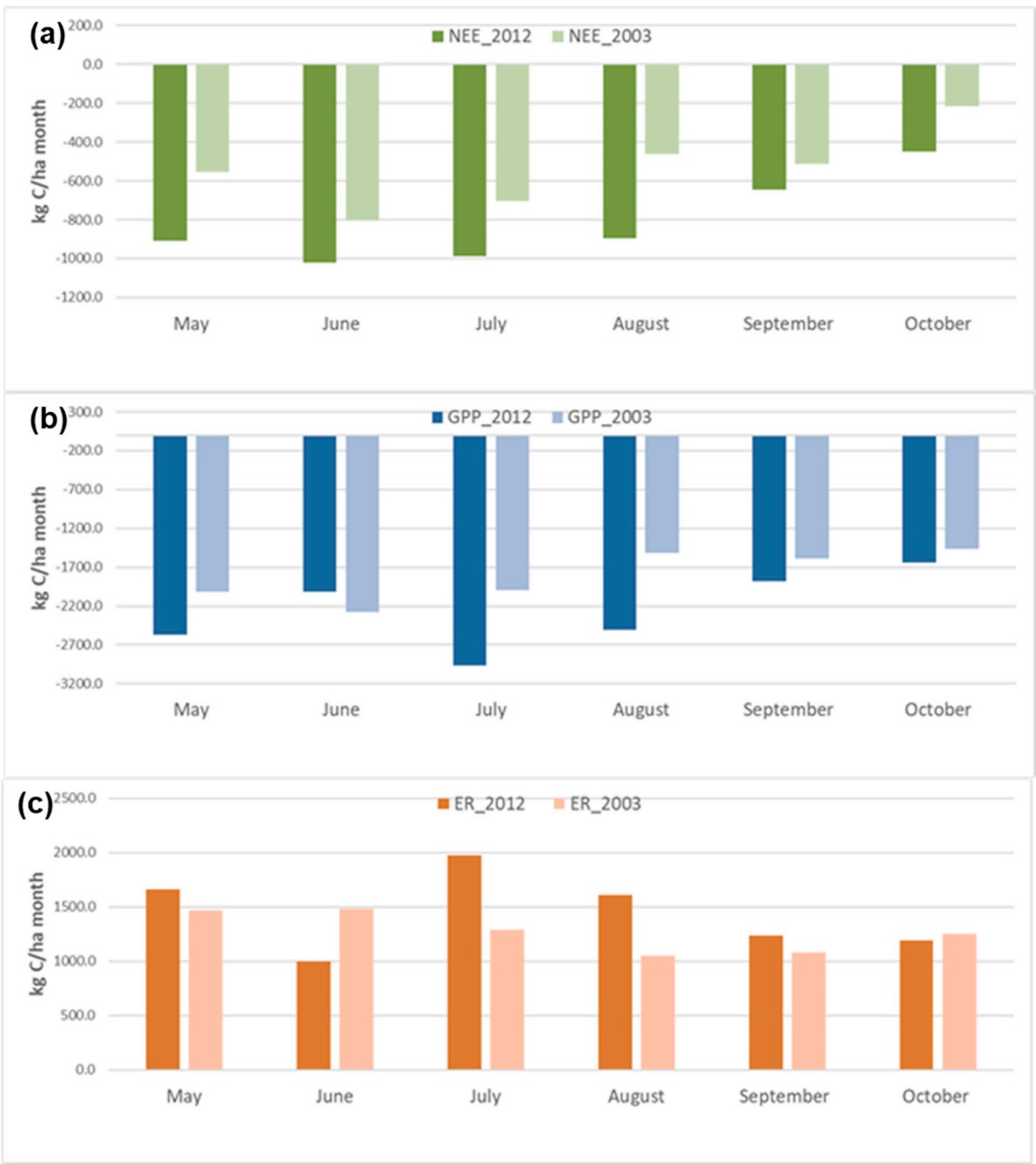

**Figure 6.** Comparison of monthly accumulated (**a**) NEE, (**b**) GPP and (**c**) RE during the vegetative season in two different years (2012 and 2003).

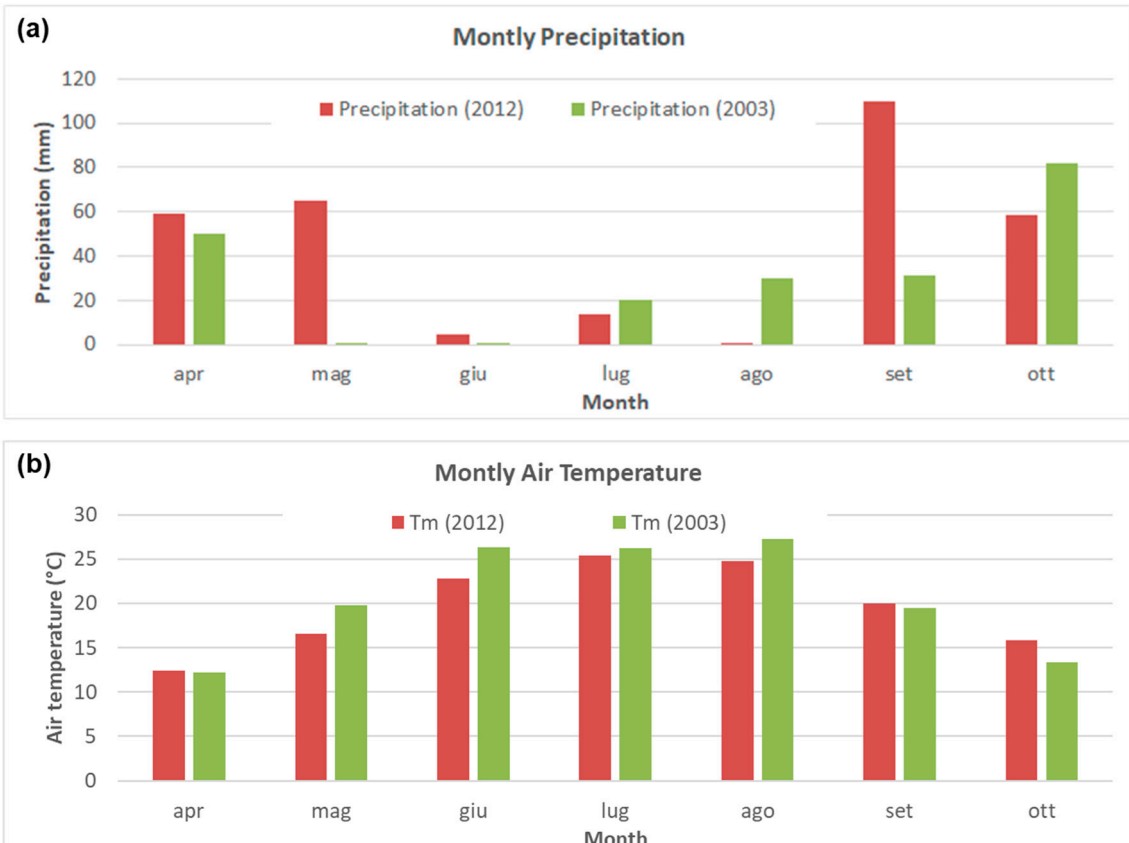

**Figure 7.** Monthly sum of (**a**) precipitation and (**b**) monthly average air temperature recorded during the vegetative season in 2012 and 2003.

### 3.2. $CO_2$-Eq Emissions from the Orchard Management

Table 3 reports the $CO_2$-eq annual emissions of the kiwi orchard management up to the farm gate.

**Table 3.** Inputs and emissions in $CO_2$ equivalent kg ha$^{-1}$ related to the orchard management, including establishment, fruit harvest and bin transport to the farm gate.

| Operation | Unit | Input | kg $CO_2$-Eq Emission |
|---|---|---|---|
| Orchard establishment | Emission yr$^{-1}$ | - | 603.76 |
| Organic fertilization | Pellet (kg) | 800 | 648.00 |
| Mineral fertilization | N (kg) | 67.5 | 633.81 |
| Foliar fertilization | Leamix (kg) | 16 | 20.79 |
| Weed control | Glyphosate (kg) | 1.08 | 54.73 |
| Fertirrigation | Bioenergy (kg) | 10 | 6.38 |
| Fertirrigation | Iron Chelate (kg) | 25 | 2.08 |
| Fertirrigation | MAP (kg) | 25 | 8.43 |
| Fertirrigation | Ammonium Nitrate (kg) | 150 | 433.50 |
| Fertirrigation | K Nitrate (kg) | 250 | 119.56 |
| Fertilization | $N_2O$ emission from soil | - | 722.51 |
| Tillage | Mowing (Number) | 4 | 145.67 |
| Tillage | Chopping/mulching (N) | 3 | 98.34 |
| Transport | Transporting of bins (N) | 80 | 78.59 |
| Transport | Moving of materials (N) | 9 | 129.70 |
| | Total (kg $CO_2$-eq) | | 3705.84 |

The sum of emissions from operations involving the use of machineries and the related combustion of fossil fuel represented 33.8% of the total (544.38 kg $CO_2$-eq ha$^{-1}$ yr$^{-1}$). Emissions from direct combustion of fossil fuels were 360.42 kg $CO_2$-eq ha$^{-1}$ yr$^{-1}$, plus an emission of 106.50 $CO_2$-eq ha$^{-1}$ yr$^{-1}$ calculated as derived from the production and distribution of diesel.

Among all the steps reported, releases of $N_2O$ from the soil, organic fertilization, mineral fertilization, and orchard establishment represented the highest values in terms of percentage (19.5%, 17.5%, 17.1%, 16.3%, respectively).

Emissions coming from fertirrigations accounted for 15.4% of the total $CO_2$-eq. In the orchard, examined here, as it generally occurs in commercial kiwifruit plantations, pruning was manual, with a very-low fuel consumption and a negligible impact.

Annual tillage operations, including transport of materials, constituted 12.2% of the total $CO_2$-eq emission.

Energy for irrigation was not directly considered in the calculation, as in the farm analyzed the distribution is carried out by gravity and the use of electric pumps is not significant. Only the emission linked to the production of tubes and their installation was calculated and included in the establishment phase. However, as irrigation is a mandatory practice requiring mass (water) and energy, a +10% correction coefficient was derived from Medici et al. [31], where kiwifruit orchards grown in a similar farming district were considered.

The overall $CO_2$-eq orchard emission resulted well in agreement with the available literature. Comparable values of $CO_2$-eq emission were found by Page et al. [3] for a typical kiwi orchard managed in New Zealand (3072 kg $CO_2$-eq ha$^{-1}$ yr$^{-1}$), where 82% of total emissions were from soil, also considering the contribution from soil biological activities and organic material decomposition, which values were derived from modeling results.

Emission data were upscaled to hectare considering the respective total yields, both in 2012 (23.9 t ha$^{-1}$) and 2003 (21.1 t ha$^{-1}$) in the two orchards located nearby and for which EC measurements had been carried out. Calculated total GHG total release from the kiwifruit management resulted equal to 4.25 t $CO_2$-eq ha$^{-1}$ yr$^{-1}$ in 2012 and 3.75 in 2003.

## 4. Discussion and Conclusions

In 2012, total GHG released from the kiwifruit orchard management, assessed by LCA, resulted in 4.25 t $CO_2$-eq ha$^{-1}$ yr$^{-1}$ while the uptake measured via EC during the active vegetation phase was as high as 4.9 t $CO_2$ ha$^{-1}$ yr$^{-1}$. This led to a positive contribution of the crop to $CO_2$ sequestration, with a 1.15 efficiency ratio (sink-source factor defined as t $CO_2$ stored/t $CO_2$-eqemitted) and a contribution to mitigation, during the average-15 years of the orchard lifetime, close to 10 t $CO_2$ ha$^{-1}$ yr$^{-1}$ of net sink, even if considering embedded upstream emissions of input.

This positive uptake value confirms and is even higher than the positive sink-source factor of 1.13 reported by Page et al. [3], as established also by the same author for an apple semi-intensive organic system.

There are several considerations reported in the literature that Hayward kiwifruits have a very efficient C fixation, where an adequate water supply is provided. NAR (Net Assimilation Rate), calculated by Laing [32] following Morgan et al. [33] and regarded as the measure of the whole kiwi plant photosynthetic rate is 560 µg $CO_2$ m$^{-2}$ s$^{-1}$, (at 650 µmol m$^{-2}$ s$^{-1}$ PPFD (Photosynthetic Photon Flux Density) and temperatures between 15 and 30 °C). Plants adapted to high PPFD conditions, as occurring in the field, rapidly adjust the sensitivity of their photosynthesis to different radiation regimes. Such adaptation allows better utilization of the incident radiation and presumably optimizes the effectiveness of making of the photosynthetic apparatus. A very efficient plant photosynthesis, concomitant with the absence of stomatal limitations has been recorded for most months in 2012, being water a not limiting factor. Several morphological characteristics of kiwi fruit vines also play a positive role in this sense: the leafy surface is very large, and reaches, as reported, values up to 35 m$^2$ vine$^{-1}$. T shaped training system limits at the



same time umbral areas and favors high PPFD at most canopy area, positively affecting whole photosynthesis of the whole orchard system.

The mitigating activity of the orchard recorded in the standard climatic year 2012, however, completely reversed under unfavorable climatic conditions recorded in 2003, when the efficiency ratio became 0.91. The occurrence of hotter, drier conditions demonstrated to be able to compromise the capability of *Actinidia* to offset the GHG emissions, also under appropriate irrigation.

The results envisage a significant role of a well-managed kiwifruit system to potentially mitigate GHGs, offering a valuable ecosystem service and a positive contribution to the objectives to reach a progressive, global reduction of C in the atmosphere. A strong dependency on the climate conditions, able to switch the system from a C sink capacity to a C source activity, has however been recorded.

The results obtained here bring a contribution to the still scarcely exploited knowledge on the complex issue of impact from, and mitigation on, climate due to different agricultural land uses and they support the quantification of complementary ecosystem services in the new vision of a "climate smart" agriculture.

In the context of net neutrality of the EU economy society, kiwi orchards have the potential to contribute with a net sink even when taking into consideration all emissions at farm gate due to its management.

**Author Contributions:** Conceptualization, F.R., C.C., N.D.V., T.G. and M.N.; methodology, F.R., C.C., N.D.V., T.G. and M.N.; software, M.N. and N.D.V.; validation, F.R., C.C., N.D.V., T.G. and M.N.; formal analysis, F.R.; investigation, F.R., T.G. and M.N.; resources, F.R., C.C., N.D.V., T.G. and M.N.; data curation, N.D.V., C.C. and M.N.; writing—original draft preparation, F.R. and M.N.; writing—review and editing, C.C. and T.G.; visualization, F.R., C.C., N.D.V., T.G. and M.N.; supervision, F.R. and M.N.; project administration, F.R.; funding acquisition, F.R. All authors have read and agreed to the published version of the manuscript.

**Funding:** This research was funded by "Programma di Sviluppo rurale dell'Emilia Romagna, 2014–2020".

**Institutional Review Board Statement:** Not applicable.

**Informed Consent Statement:** Not applicable.

**Data Availability Statement:** Not applicable.

**Acknowledgments:** Authors are grateful to the owner of the farm, for hosting the Eddy Covariance tower for long time in his fields, and for having so kindly answered to our long and detailed questions on farm management. Thanks are due to Stefano Anconelli and Paolo Mannini, Consorzio di Bonifica del Canale Emiliano Romagnolo ER.

**Conflicts of Interest:** The authors declare no conflict of interest.

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
