# Peer review of "Is Agriculture Always a GHG Emitter? A Combination of Eddy Covariance and Life Cycle Assessment Approaches to Calculate C Intake and Uptake in a Kiwifruit Orchard"

_sustainability, doi:10.3390/su13126906_

Round 1

Reviewer 1 Report

Conclusions must be improved.

Study background need to improved.

Add extensive detail of previous research gaps

Author Response

Sentence included in the introduction (pag 2 row 86) for study background and previous research gaps.

Conclusions has been slightly revised as it was English.

Reviewer 2 Report

All figures have low quality, needs to improve...Fig 4 I can't read the legend, and the caption do not explain the legend.

Author Response

The figure 4 was modified to make it readable.

The other pictures have been enlarged when possible.

Reviewer 3 Report

Interesting paper that addresses the emissions of crop cycles of known fruit species. Well organized and described, some insights into the data relating to the use of the machines, number of interventions, type of management, powers used would be useful. Another rather neglected aspect concerns the material for diseases and pests, the irrigation and above all its management and methods of administration, naturally in a manner capable of providing the imprint that this technology entails in the production cycle. Varietal comparisons should always be carried out in absolute parity of the management system. 

Author Response

Thank you for the comments. We have assumed a conventional management and in our opinion the inclusion of specific data on pest and disease treatments, irrigation and others did not match the intention of the paper. In the sense that the orchard was grown under optimal conditions and carbon dynamics would not have been modified by minor differences in management.

We fully understand the point about varietal comparison, but it was not possible to measure a similar surface of a different variety due to the cost of the instrumentation and the complexity of measurements.

We believe that data obtained can be scaled to other kiwifruit varieties.